# Relationship between Glucagon-like Peptide-1 Receptor Agonists and Cardiovascular Disease in Chronic Respiratory Disease and Diabetes

**DOI:** 10.3390/biomedicines12030488

**Published:** 2024-02-22

**Authors:** Jun-Jun Yeh, Chih-Chien Li, Chang-Wen Tan, Chia-Hsun Li, Tung-Han Tsai, Chia-Hung Kao

**Affiliations:** 1Department of Thoracic Medicine, Family Medicine, Geriatric Medicine and Medical Research, Ditmanson Medical Foundation, Chia-Yi Christian Hospital, Chia-Yi 600, Taiwan; anvin.funlan@msa.hinet.net; 2Department of Psychotherapy, Clinical Psychology Center, Chia-Yi Christian Hospital, Chia-Yi 600, Taiwan; 07239cych@gmail.com; 3Department of Family Medicine and Medical Research, Ditmanson Medical Foundation, Chia-Yi Christian Hospital, Chia-Yi 600, Taiwan; 07763cych@gmail.com (C.-W.T.); kophylit@gmail.com (C.-H.L.); donny85050245@gmail.com (T.-H.T.); 4Graduate Institute of Clinical Medical Science, College of Medicine, China Medical University, Taichung 404, Taiwan; 5Department of Nuclear Medicine, PET Center, China Medical University Hospital, Taichung 404, Taiwan; 6Artificial Intelligence Center, China Medical University Hospital, Taichung 404, Taiwan; 7Department of Bioinformatics and Medical Engineering, Asia University, Taichung 413, Taiwan

**Keywords:** glucagon-like peptide-1 receptor agonists (GLP-1RAs), stroke, heart disease, COVID-19

## Abstract

The purpose of this paper is to assess the effect of glucagon-like peptide-1 receptor agonists (GLP-1RAs) on stroke or heart disease in patients having chronic respiratory disease and diabetes (CD) with underlying diseases related to COVID-19. From 1998 to 2019, we adjusted competing risk by assessing the effect of GLP-1RAs on stroke or heart disease in a CD cohort after propensity matching based on the Taiwan National Health Insurance Research Database. We also used the time-dependent method to examine the results. GLP-1 RA and non-GLP-1 RA user groups included 15,801 patients (53% women and 46% men with a mean age of 52.6 ± 12.8 years). The time between the diagnoses of DM and the initial use of the GLP-1 RA among the stroke subcohort (<2000 days) was shorter than that of the heart disease subcohort (>2000 days) (all *p*-values < 0.05). The overall risks of stroke, ischemic, and hemorrhagic stroke were significantly lower in GLP-1 RA users than nonusers. The adjusted subhazard ratio (aSHR) was 0.76 [95% CI 0.65–0.90], 0.77 [95% CI 0.64–0.92], and 0.69 [95% CI 0.54–0.88] (*p* < 0.05 for all). Furthermore, a ≥351-day use had a significantly lower stroke risk than GLP-1 RA nonusers (aSHR 0.35 [95% CI 0.26–0.49]). The time-dependent method revealed the same result, such as lower stroke, and ischemic or hemorrhagic stroke risk. In contrast, the cardiac arrhythmia incidence was higher in GLP-1 RA users with an aSHR of 1.36 [95% CI 1.16–1.59]. However, this risk disappeared after the ≥351-day use with 1.21 (0.98, 1.68) aSHR. Longer GLP-1 RA use was associated with a decreased risk of ischemic or hemorrhagic stroke and the risk of cardiac arrhythmia disappears in a CD cohort. Both a shorter lag time use of the GLP-1 RA and a longer time use of GLP-1 RA were associated with a decreased risk of ischemic or hemorrhagic stroke in the CD cohort. The GLP-1 RA use in the early stage and optimal time use in the CD cohort may avoid the stroke risk.

## 1. Introduction

The prevalence of chronic respiratory diseases (CRDs) is estimated to have increased by 40% between 1990 and 2017. They are among the leading causes of morbidity and mortality worldwide. Meanwhile, the CRD patients have an impact on the outcomes of coronavirus disease 2019 (COVID-19) [1,2,3]. CRDs are airway and other lung structure diseases, including chronic obstructive pulmonary disease (COPD), asthma, lung cancer, tuberculosis, bronchiectasis, interstitial lung disease (ILD), sleep apnea, sarcoidosis, pulmonary hypertension, and chronic respiratory failure [4]. CRDs may interact with diabetes mellitus (DM) and cardiovascular diseases (CVD). For example, DM has an increased risk of tuberculosis, and both are associated with a higher risk of lung cancer. In COPD, ILD increases the risk for DM and CVD [5,6]. Thus, CRDs may link DM and CVD, coexist, and interact, even in young adults in the COVID-19 era [5,7,8].

Glucagon-like peptide-1 receptor agonists (GLP-1 RAs) have pleiotropic effects, including preventing the development and progression of coronary atherosclerosis, epicardial coronary artery vasospasms, and structural/functional changes in coronary microvasculature. They can also decrease systolic blood pressure, reduce low-density lipoprotein cholesterol, and promote body weight loss. GLP-I RA is expressed in multiple organs, including the brain, kidney, stomach, heart, and predominantly lung epithelia and immune cells. Moreover, they provide multiple benefits, such as controlling lung injury caused by inflammation through pulmonary protective effects, anti-obesogenic properties, and modulation of gut microbiota. Altogether, GLP-I RA may improve COVID-19 outcomes because they avoid the development of ischemic heart disease or stroke in DM patients with atherosclerosis [9,10].

During the COVID-19 pandemic, CRD and DM (CD) were associated with COVID-19 and these diseases were also predisposing factors for CVD [1,2]. Moreover, long COVID-19 may exacerbate the severity of the CVD in patients with CD [11,12]. GLP-I RAs clinical trials and cardiovascular outcome trials had considerable differences in design and enrollment, limiting comparisons between them [13,14]. In a meta-analysis of GLP-1 RAs, they reduce nonfatal stroke risk in DM patients [15]. A Taiwan study revealed that longer use and a higher dose of GLP-I RAs decrease the ischemic stroke risk in patients with DM, without established atherosclerotic disease (ACVD) [16]. However, Wei et al. report that treatment with GLP-I RA in patients with DM has no significant effect on the risk of major arrhythmias but significantly reduces the hypoglycemia and pneumonia risks [17]. Meanwhile, the benefit of GLP-I RA in heart diseases such as heart failure or acute myocardial infarction in DM is debatable [18,19,20]. Moreover, we did not find a relationship between GLP-1 RA and CVD in CD patients who did not have established ACVD [21,22].

The aim of this study is that we would like to clarify the impact of the GLP-1 RA on the risk of stroke and heart disease among the CD cohort. Therefore, we addressed this topic based on a time-dependent analysis after propensity matching of the study population.

## 2. Methods and Materials

All individual patient records were anonymized before access according to the ethical guidelines of the Taiwan National Health Insurance Research Database (NHIRD); therefore, the requirements for informed consent were waived by the Research Ethics Committee. The Institutional Review Board and Ethics Committee of China Medical University approved the protocol (IRB approval number: No.CMUH104-REC2-115[CR-7]).

### 2.1. Operational Exposure and Outcome Definitions (Appendix A)

#### Cohort Identification

***Patients with DM***: newly diagnosed patients were identified between 1998 and 2018 if they had (1) ≥1 inpatient record with a DM diagnosis (International Classification of Diseases, Ninth Revision, Clinical Modification (ICD-9-CM): 250; ICD-10-CM: E08–E13) or ≥2 outpatient department (OPD) records with a DM diagnosis within the given years, or (2) ≥1 OPD record with a DM diagnosis and prescription records of glucose-lowering drugs within the given years.

***Patients**with**CRD**having an impact on the outcomes of**COVID-19***: newly diagnosed patients were identified between 1998 and 2018 if they had chronic respiratory diseases, such as COPD, asthma, lung cancer, bronchiectasis, tuberculosis, ILD, sleep apnea, sarcoidosis, pulmonary hypertension, and chronic respiratory failure with ≥1 inpatient or ≥2 OPD records [1,2].

***Definition of CD cohort***: the initial cohort included patients with new-onset CRD and DM between 1998 and 2018.

***Inclusion criteria:*** if a patient had at least these two disease diagnosis requirements within 1 year of follow-up, a washout period of a minimum of 3 years was used to ensure that the patient was newly diagnosed with CD.

### 2.2. Exclusion Criteria

We excluded patients (1) whose sex was unknown, were not Taiwanese citizens, or younger than 20 years; (2) had no CRD-related prescription requirements or related management or procedure 1 year after the first CRD diagnosis; (3) had a disease history of CRD; (4) were diagnosed with DM before the first CRD diagnosis or no antidiabetic prescription statement or DM-related management or procedure, and (5) received GLP-1 RAs < 3 times. The subsequent exclusion was used to increase the homogeneity of the study population.

### 2.3. Definition of GLP-1 RA Use

The index date indicates when GLP-1 RA was first prescribed. The length of exposure to CD treatment with GLP-1 RA therapy was calculated as the time from the index date to the occurrence of the study outcomes or the final GLP-1 RA prescription during the observation period. A 90-day interval between prescription refills indicated therapy discontinuation. Patients were eliminated from the study when GLP-1 RA treatment was discontinued, a treatment switch occurred between the use of GLP-1 RA, or patients withdrew from the National Health Insurance program. Patient follow-up examinations continued until the investigated outcomes or the end of the study (31 December 2019), whichever occurred first.

### 2.4. Outcomes

The primary outcomes were stroke, ischemic stroke, or hemorrhagic stroke. The secondary outcomes were cardiac arrhythmia, CAD, and heart failure hospitalization. All-cause mortality was confirmed by linking our study dataset with the national electronic death registry.

### 2.5. Definitions of Covariates

#### 2.5.1. Comorbidities Having Impact on the Outcomes of COVID-19

All study covariates were defined according to the ICD-9-CM code (≥2 diagnoses for OPD or ≥1 diagnosis during hospitalization) within 1 year of the index date. The covariates included comorbidities associated with COVID-19 such as hypertension, hyperlipidemia, gout, tobacco dependence-related, chronic renal disease, venous thrombosis, and depression or substance-related disease [1,2,11,12]. The Appendix A shows full names of ICD-9-CM or ICD-10-CM diagnoses.

#### 2.5.2. Medications and Psychotherapy Having Impact on the Outcomes of COVID-19

Medications were defined according to the Anatomical Therapeutic Chemical classification. The medications included AGI, metformin, insulin, DPP4, meglitinides, TZD, sulphonylurea, statins, antidepressants, antihypertensives, and antithrombotics. Meanwhile, psychotherapy entered into the analysis. The Appendix A shows the full names of medications and psychotherapy.

### 2.6. Statistical Analysis

#### Propensity Score Matching

The demographic characteristics and the prevalence of comorbidities in GLP-1 RA and non-GLP-1 RA treatment groups were compared using the *χ*^2^ test for categorical variables and a *t*-test for continuous variables. The incidence densities by sex, age, comorbidity, medications, and psychotherapy were also calculated for each cohort. The number of stroke patients diagnosed annually was calculated chronologically from 1998 to 2018. We conducted a propensity score-matched analysis to minimize the effects caused by confounding factors. We calculated the propensity score for GLP-1 RA usage likelihood via multivariable logistic regression analysis, which was conditional on the baseline covariates listed in Table 1.

### 2.7. Time-Dependent Analysis

Since patients with CD may not have taken drugs regularly during the study period, which may lead to an overestimation of the drug’s effect, we used the Cox proportional hazard model with time-dependent exposure covariates to estimate the stroke risk to reduce this bias.

### 2.8. Fine and Gray Model for Competing Risk

After accounting for competing risks of death or withdrawal from NHI, we used the Fine and Gray model, which extends the standard Cox proportional hazard regression model, to estimate the cumulative incidence of stroke. Death events were identified based on hospital discharge due to death and withdrawal from NHI, as indicated in NHIRD. Univariate and multivariate competing-risks regression models were proposed to calculate subhazard ratios (SHRs), adjusted SHR (aSHR), and 95% CIs of the stroke risk associated with GLP-1 RA. The multivariate models were simultaneously adjusted for demographic characteristics, baseline comorbidities, medications, and psychotherapy.

All statistical analyses were performed using the SAS statistical software package, version 9.1 (SAS Institute, Cary, NC, USA). The cumulative incidence survival curve was plotted using Stata version 11.1 (Stata LP, College Station, TX, USA). Results with a two-tailed *p* < 0.05 were considered statistically significant.

## 3. Results

The demographic characteristics of the study population are presented in Table 1.

After Propensity Score Matching, GLP-1 RA (n = 15,801) and non-GLP-1 RA (n = 15,801) cohorts were identified. The age distribution between the GLP-1 RA cohort and non-GLP-1 RA cohort is different *(p* = 0.03). The frequency of older adults with GLP-1 RA use is lower than non-GLP-1 RA cohort (18.4% vs. 19.0%). Compared with the non-GLP-1 RA cohort, sex, comorbidities, medication, and psychological therapy were not significantly different except for the GLP-1 RA use cohort that had the lowest frequency of hyperlipidemia, metformin, and sulphonylurea use and higher use of meglitinides, antiarrhythmic drugs, and antidepressants (Table 1).

Table 2 displays the incidence of stroke and heart disease by the Cox model measured hazards ratio according to medication status such as the time between the diagnoses of the DM and the initial use of the GLP-1 RA.

The logistic regression models were adjusted for potential confounders, such as age, sex, comorbidities, and medication. The adjusted subhazard ratio (aSHR) revealed that GLP-1 RA use had a lower aSHR for stroke, ischemic stroke, hemorrhagic stroke, a neutral effect on CAD and heart failure, and a higher aSHR for arrhythmia (Figure 1 and Figure 2, Table 2). The CD cohort with the arrhythmia had the longest and hemorrhagic stroke had the shortest lag time between the DM diagnosis and initial GLP-1 RA use (Table 2, footnote).

Table 3 displays the comorbidities, medications, and psychological therapy that might interfere with the effect of GLP-1 RA use for stroke and heart diseases.

Patients with AGI, thrombotic agents, and psychological therapy exposure had an additive effect in attenuating the stroke risk among GLP-1 RA users. Meanwhile, GLP-1 RA users had a lower aSHR, even with comorbidities. However, GLP-1 RA use with comorbidities had a higher risk of arrhythmia (Table 3, Appendix A, Figure 3).

Table 4 displays the incidence and adjusted subhazard ratio of stroke after stratification of the cumulative use day of GLP-1 R A therapy.

The GLP-1 RA use still had a significant dose-response effect for stroke, especially with ≥351 days of use (0.35 [0.26, 0.49], *p* < 0.001) (Table 4).

Table 5 displays the time-dependent regression for the GLP-1 RA use.

The overall estimated SHRs in the GLP-1 RA use cohort had a lower aSHR for stroke, ischemic stroke, and hemorrhagic stroke (Table 5).

Figure 1 display the stroke and arrhythmia cumulative incidence curve.

The GLP-1 RA use cohort had a lower risk of ischemic stroke, and hemorrhagic stroke than the non-GLP-1 RA use cohort during the observation periods (Figure 1 and Figure 2). In contrast, GLP-1 RA use cohort had a higher risk of cardiac arrhythmia (Figure 3).

## 4. Discussion

### 4.1. Summary of the Main Results

This is the first study that demonstrates a correlation between stroke and CD, two related medical conditions. Patients with GLP-1 RA in our study carried a lower risk of developing stroke (cHR = 0.71(0.6–0.84). After conducting a multivariate analysis, the GLP-1 RA cohort still had a significantly lower risk of stroke with an adjusted HR of 0.76 (95% 0.65–0.90). After conducting a competing-risk model analysis, the GLP-1 RA cohort with the longest time of use (≥351-day) had the lowest aSHR0.42 (0.32–0.55) [23,24]. Our findings have major implications due to the financial burden caused by CD treatment worldwide. First, clinicians should actively monitor predisposing factors such as hypertension, hyperlipidemia, and gout for stroke in patients with a history of hospitalization due to CD. Second, growing evidence shows that the incidence of coronavirus and CD, such as ILD, with DM, has increased. Thus, public health officials should promote long-term GLP-1 RA treatment among high-risk patients and a more broadly effective education team for metabolic syndrome.

### 4.2. An Explanation of Findings

The control group patients were matched by age, sex, comorbidities, and medication after the propensity score to control the possible confounding factors affecting stroke or GLP-1R A. Two cohorts, cases and comparisons, were homogeneous regarding sex, comorbidities, and medications, except for age, hyperlipidemia, metformin, meglitinides, sulphonylurea, antiarrhythmics, and antidepressants.

The natural outcomes of CD include elevated cardiovascular disease risk and stroke. One question that might be raised is whether older adults should be considered a competing risk for stroke. However, based on these studies, the range of elderly outcomes is difficult to evaluate because respiratory failure or a tumor might lead to early death without experiencing cardiovascular disease or stroke. In addition, the extent to which CD marks or causes an elevated stroke among elderly adults cannot be determined with observational data. Thus, our study found it difficult to eliminate primary CRD and its complications as a confounding factor for stroke.

Although we did not find a significant effect on the risk of stroke in the elderly subgroup with GLP-1 RA use, this result remains uninterpretable per current knowledge. Due to NHIRD policies and injection and high-cost inconveniences, early use and generalizability of GLP-I RA remain debatable. Thus, the delayed use of GLP-I RA may be a factor for the null effect on stroke risk in older adults [24]. Meanwhile, adverse reactions to GLP-I RA may be associated with poor adherence in the elderly [24] and, moreover, owing to only the long-time use (≥351 days) with a lower stroke risk. Therefore, these elderly individuals did not have adequate time to attenuate stroke risk, and a null effect for stroke risk was found. GLP-I RA users aged > 65 had a longer lag time for initial use, and the frequency of GLP-I RA use (18.4%) was low [23]. Moreover, the frequency of the long-time GLP-I RA use (≥351 days) was also low (10%). These findings in our results support our speculations (Table 1, Table 3, footnote).

In the cardiac arrhythmia subgroup, the increased adjusted HRs should be interpreted cautiously because GLP-1 RA use was not associated with the risk of other heart disease, coronary artery disease, and heart failure. To reveal the mechanisms of persistent cardiovascular sequelae following CD, we reviewed all relevant studies and proposed that CD causes the initial cardiovascular and cerebrovascular injury, and the host immune response and additional underlying comorbid diseases may contribute to secondary cardiovascular and cerebrovascular insults [25].

First, CDs occur more commonly in each other’s presence. They are associated with a similar inflammatory pathophysiology, suggesting that a common process results in the clinical overlap of CD. Interleukin-6 (IL-6) and TNF-α appear to be central mediators in this process. This finding suggests that factors influencing their production may lead to a cascade of events, making CRD, DM, cardiovascular, and cerebrovascular coexistence and interplay more likely. Second, IL-6 level is in line with CRD, such as COPD, with exacerbations with hyperglycemia; thus, a higher IL-6 level was found in CD. This overexpression of IL-6 may contribute to cardiovascular diseases, such as CAD, heart failure, and arrhythmias [26,27,28,29]. Thus, IL-6-related oxidative stress of atherosclerosis counterbalances the antiatherosclerosis effect of GLP-1RA, leading to an aHR of CAD and heart failure with a null effect [30].

An association of GLP-I RA with arrhythmia was debated in a study by Fauchier et al. [31,32]. GLP-I RA was associated with a higher risk of arrhythmia in this study [32]. Hyperlipidemia accompanied by a high IL-6 level was a predisposing factor for arrhythmia. Hyperlipidemia and the primary CD effect on atherosclerosis may overwhelm the antiatherosclerosis of GLP-I RA, leading to a higher risk of cardiac arrhythmia. Another explanation is that GLP-I RA use with SA node stimulation may contribute to arrhythmia in the CD cohort [33]. Patients with short-term GLP-1 RA (<351 days) use and a higher risk of arrhythmia may support this hypothesis. However, when usage time increases, the antiatherosclerosis effect of GLP-I RA may counterbalance the effect on arrhythmia; thus, the higher aSHR disappears after ≥351 days of use. The null effect on arrhythmias in long-term use (≥351 days) was also found in this study (Appendix A). These heterogeneous results warrant further study.

Hypoglycemia, postprandial hyperglycemia, and glucose fluctuations were associated with an accelerated risk of stroke [34]. The α-glucosidase inhibitor (AGI) may delay carbohydrate absorption, reducing postprandial hyperglycemia. Meanwhile, the remaining unabsorbed nutrients might increase L-cell activity and facilitate incretin secretion for glycemic control, dampening the postprandial plasma glucose spike [35]. AGI use in the DM cohort may be rare with hypoglycemia. This combination formula (AGI and GLP-1 RA) may avoid glucose fluctuations, leading to a lower risk of stroke [36]. The additive effect of AGI and GLP-1 RA use for stroke prevention was found in this study. In a previous study, the use of AGI was associated with a significantly lower risk of hospitalizations due to major atherosclerotic events, ischemic stroke, and hypoglycemia, in line with our results [37]. Similar to previous studies, psychological therapy may attenuate the risk of glucose fluctuations, leading to a lower stroke risk [38].

In summary, both a short lag time (<2000 days) use and longer time (≥351 days) use of GLP-1 RA were associated with a lower risk of stroke in the CD cohort. The neutral effect on cardiovascular disease, such as CAD, heart failure, or a higher aHR with cardiac arrhythmia, was found during GLP-I RA usage. However, the decreasing trend of the value of aSHR for cardiac arrhythmia in long-term use (≥351 days) was also found in this study. When interpreting these findings, we must consider the primary CD effect or its comorbidities effect or lag time or duration of GLP-1 RA usage on developing these opposite results.

## 5. Strengths

First, this is the first time-dependent study examining stroke risk effects in the CD cohort with underlying diseases related to COVID-19. A time-dependent Cox regression model eliminates guarantee-time bias by using drug usage as a time-dependent covariate. This model has the advantage of using all study follow-up data since it starts the analysis at the time of cohort entry. Second, the Kaplan–Meier method overestimated the incidence of both events. A relative factor of 39% overestimated stroke incidence in a previous study [39]. The Fine–Gray model with aSHR demonstrated better calibration than the Cox model, which consistently overpredicted stroke incidence.

## 6. Limitations

Our findings should be interpreted in the context of the inherent limitations of an observational study using administrative databases. First, the analysis did not contain detailed information regarding smoking or drinking habits, socioeconomic status, and family history of cardiovascular disease. The relevant clinical variables were unavailable, including blood pressure, glucose level, and carotid duplex sonography or brain computed tomography as important stroke severity measures with prognostic implications. These variables could not be adjusted during analysis. However, the claimed data regarding the diagnoses of stroke, heart disease, and comorbidities, including chronic renal disease from the NHIRD, were nonetheless reliable. Second, the evidence derived from a retrospective cohort study has generally lower statistical quality than that derived from randomized trials because of potential biases. However, we matched controls with PS to balance the demographic characteristics of GLP-1 RA users’ and nonusers’ cohorts, and multivariate analysis was used to define the robustness of our results bias, resulting from residual confounders that might have affected the results. Another limitation of this study is the limited generalizability of the findings. Although we used NHIRD data to obtain an adequate sample size, our results cannot be generalized to all GLP-I RA users because we recruited only patients living in Taiwan. Therefore, large-scale, multi-center research is warranted to confirm our results.

## 7. Conclusions

Both a shorter lag time use of the GLP-1 RA and longer use of GLP-1 RA were associated with a decreased risk of ischemic or hemorrhagic stroke in the CD cohort. The GLP-1 RA use in the early stage with optimal time use in the CD cohort may avoid the risk of stroke. These findings suggest implications for clinical practice and further research.

## Figures and Tables

**Figure 1 biomedicines-12-00488-f001:**
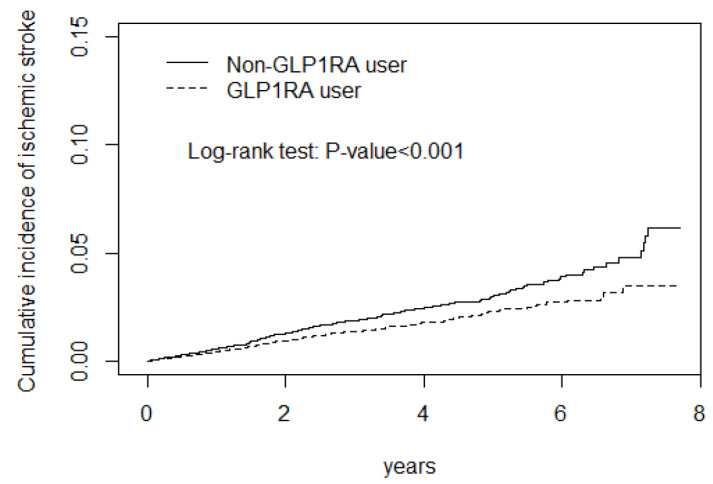
GLP-1RA use cohort had a lower risk of ischemic stroke than non-GLP-1RA use cohort during the observation periods.

**Figure 2 biomedicines-12-00488-f002:**
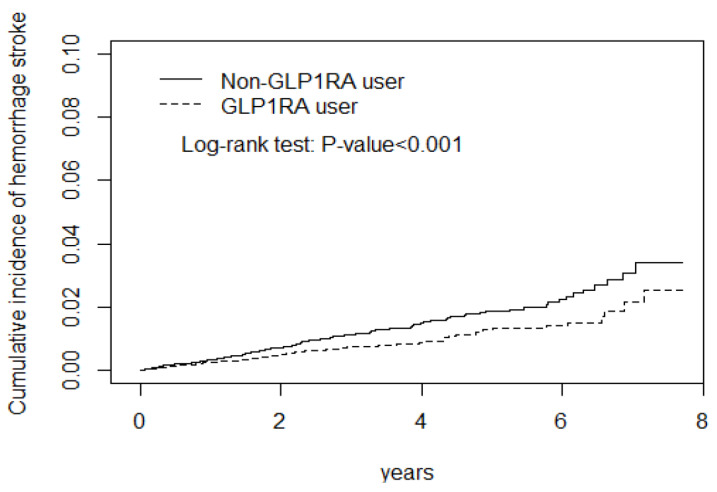
GLP-1 RA use cohort had a lower risk of hemorrhagic stroke than non-GLP-1 RA use cohort during the observation periods.

**Figure 3 biomedicines-12-00488-f003:**
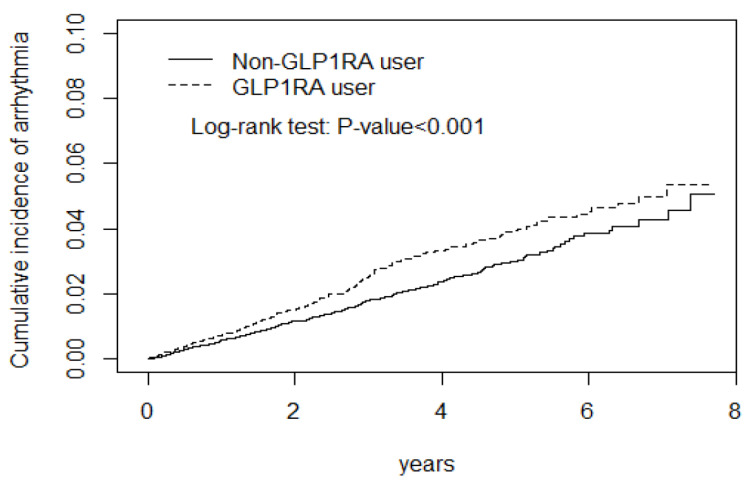
GLP-1RA use cohort had a higher risk of cardiac arrhythmia.

**Table 1 biomedicines-12-00488-t001:** Distributions of demographic and clinical comorbid status in GLP-1 RA among diabetes with chronic respiratory disease (CRD) by Propensity Score Matching.

	GLP-1 RA				
	No N = 15,801		Yes N = 15,801		
	n	%	n	%	*p*-Value
**Age, years**					0.03 *
≤49	6153	38.9	6373	40.3	
50–64	6640	42.0	6521	41.3	
≥65	3008	19.0	2907	18.4	
Mean ± SD ^a^	54.7	12.7	52.6	12.8	0.00 *
**Gender**					0.57
Women	8481	53.7	8531	54.0	
Men	7320	46.3	7270	46.0	
**Comorbidity**					
Hypertension	11,013	69.7	10,957	69.3	0.49
Hyperlipidemia	14,388	91.7	14,376	91.0	0.03 *
Chronic renal disease	1861	11.8	1977	12.5	0.05
Gout	2430	15.4	2479	15.7	0.45
Tobacco dependence related	341	2.16	372	2.35	0.24
Venous thrombosis	19	0.12	22	0.14	0.64
Depression or substance-related disease	1927	12.2	1990	12.6	0.28
**Medications**					
AGI	8856	56.1	8926	56.5	0.43
Metformin	15,737	99.6	15,660	99.1	0.00 *
Insulin	13,008	82.3	12,972	82.1	0.60
DPP-4 inhibitor	13,973	88.4	13,873	87.8	0.08
Meglitinides	4878	30.9	5072	32.1	0.02 *
TZD	9872	62.5	9813	62.1	0.49
Sulphonylurea	14,731	93.2	14,503	91.8	0.00 *
Statin	13,727	86.9	13,637	86.3	0.14
Antidepressants	5311	33.6	5494	34.8	0.03 *
Antihypertensive	13,813	87.4	13,725	86.9	0.14
Antithrombotic	7428	47.0	7312	46.3	0.19
**Psychotherapy**	5300	33.5	5420	34.3	0.06

AGI: alpha glucosidase inhibitor, DPP-4 inhibitor: dipepidyl peptidase 4 inhibitor, TZD: thiazolidinedione. Chi-square test, ^a^
*t*-test. Abbreviations: * *p* value < 0.05.

**Table 2 biomedicines-12-00488-t002:** Incidence of stroke and heart disease by Cox model measured hazard ratio according to medication status.

	GLP-1 RA							
	No			Yes				
Outcome	Event	PY	Rate ^#^	Event	PY	Rate ^#^	Crude SHR (95% CI)	Adjusted SHR ^a^ (95% CI)
All ^b^	1285	43,760	29.4	1251	43,012	29.1	1.00(0.92, 1.08)	1.03(0.95, 1.11)
Coronary ^c^ artery disease	647	44,742	14.5	667	43,990	15.2	1.06(0.95, 1.18)	1.09(0.97, 1.21)
Arrhythmia ^d^	271	45,452	5.96	354	44,608	7.94	1.34(1.15, 1.57) ***	1.36(1.16, 1.59) ***
Stroke ^e^	354	45,756	7.74	239	45,163	5.29	0.71(0.60, 0.84) ***	0.76(0.65, 0.90) **
Ischemic ^f^ stroke	291	45,398	6.41	207	44,956	4.60	0.73(0.61, 0.87) ***	0.77(0.64, 0.92) **
Hemorrhagic stroke ^g^	172	45,677	3.77	111	45,135	2.46	0.66(0.52, 0.84) ***	0.69(0.54, 0.88) **
Heart failure ^h^	304	45,450	6.69	307	44,799	6.85	1.04(0.88, 1.21)	1.09(0.93, 1.28)

SHR = Subhazard Ratio; CI = Confidence Interval; PY = Persona Year; Rate ^#^, incidence rate, per 1000 person-years; Crude SHR, Relative Subhazard Ratio. ^a^ Adjusting for age, gender, comorbidities of hypertension, hyperlipidemia, chronic renal disease, gout, tobacco dependence-related, venous thrombosis, depression or substance-related, and medications of alpha glucosidase inhibitor (AGI), metformin, insulin, dipepidyl peptidase 4 (DPP-4) inhibitor, meglitinides, thiazolidinedione (TZD), sulphonylurea, statin, antidepressants, antihypertensive, antithrombotic, and psychological therapy; ** *p* < 0.01, *** *p* < 0.001. The time between the diagnoses of the DM and the initial use of the GLP-1 RA; ^b^ (2152 ± 7.7) days, ^c^ (2259 ± 9.8) days, ^d^ (2681 ± 9.2) days, ^e^ (1789 ± 4.3) days, ^f^ (1819 ± 1.2) days, ^g^ (1687 ± 3.5) days, ^h^ (2018 ± 6.4) days. Comparison of the (e, f, g < 2000-day) with (b, c, d, h > 2000-day), all *p*-values < 0.00.

**Table 3 biomedicines-12-00488-t003:** Incidence of stroke by age, sex, comorbidity, and medication and Cox model measured hazards ratio according to medication status.

	GLP-1 RA							
	No			Yes				
Variables	Event	PY	Rate ^#^	Event	PY	Rate ^#^	Crude SHR (95% CI)	Adjusted SHR ^a^ (95% CI)
Age, years								
≤49 ^♥^	81	19,420	4.17	50	19,705	2.54	0.62(0.44, 0.89) **	0.65(0.46, 0.92) *
50–64 ^♥♥^	154	19,049	8.08	106	18,399	5.76	0.75(0.59, 0.96) *	0.78(0.61, 0.99) *
≥65 ^♥♥♥^	119	7287	16.3	83	7059	11.8	0.75(0.57, 0.99) *	0.79(0.59, 1.06)
p for interaction								0.43
Gender								
Women	175	25,053	6.99	118	24,944	4.73	0.71(0.56, 0.89) **	0.76(0.60, 0.96) *
Men	179	20,702	8.65	121	20,219	5.98	0.71(0.57, 0.90) **	0.76(0.60, 0.96) *
p for interaction								0.85
Comorbidity ^‡^								
No	4	1347	2.97	0	1133	0		
Yes	350	44,408	7.88	239	44,030	5.43	0.71(0.61, 0.84) ***	0.77(0.65, 0.90) **
p for interaction								0.00
Medications								
AGI								
No	118	19,580	6.03	98	19,069	5.14	0.89(0.68, 1.17)	0.96(0.73, 1.26)
Yes	236	26,175	9.02	141	26,094	5.40	0.62(0.50, 0.77) ***	0.67(0.54, 0.82) ***
p for interaction								0.04
Metformin								
No	3	185	16.2	3	340	8.83	0.64(0.13, 3.20)	0.30(0.02, 5.69)
Yes	351	45,570	7.70	236	44,823	5.27	0.71(0.60, 0.84) ***	0.76(0.65, 0.90) **
p for interaction								0.92
Insulin								
No	17	7609	2.23	12	7437	1.61	0.75(0.36, 1.56)	0.85(0.41, 0.76)
Yes	337	38,147	8.83	227	37,726	6.02	0.70(0.60, 0.83) ***	0.76(0.64, 0.90) **
p for interaction								0.83
DPP-4 inhibitor								
No	31	5115	6.06	26	5245	4.96	0.85(0.50, 1.42)	0.91(0.53, 1.54)
Yes	323	40,641	7.95	213	39,918	5.34	0.70(0.59, 0.83) ***	0.75(0.63, 0.89) **
p for interaction								0.47
Meglitinides								
No	215	31,478	6.83	140	30,617	4.57	0.69(0.56, 0.85) ***	0.73(0.59, 0.90) **
Yes	139	14,277	9.74	99	14,546	6.81	0.73(0.57, 0.95) *	0.79(0.61, 1.02)
p for interaction								0.67
TZD								
No	121	16,635	7.27	77	16,470	4.68	0.68(0.51, 0.90) **	0.75(0.56, 1.00) *
Yes	233	29,121	8.00	162	28,693	5.65	0.73(0.60, 0.89) **	0.77(0.63, 0.94) **
p for interaction								0.63
Sulphonylurea								
No	13	2864	4.54	11	3336	3.30	0.75(0.34, 1.66)	0.70(0.30, 1.64)
Yes	341	42,892	7.95	228	41,827	5.45	0.71(0.60, 0.84) ***	0.76(0.64, 0.90) **
p for interaction								0.95
Statin								
No	30	5743	5.22	24	5957	4.03	0.78(0.46, 1.33)	0.85(0.49, 1.47)
Yes	324	40,012	8.10	215	39,206	5.48	0.70(0.59, 0.83) ***	0.75(0.63, 0.89) **
p for interaction								0.68
Antidepressants								
No	215	30,269	7.10	129	29,424	4.38	0.65(0.52, 0.80) ***	0.69(0.55, 0.86) ***
Yes	139	15,487	8.98	110	15,739	6.99	0.80(0.63, 1.03)	0.86(0.67, 1.11)
p for interaction								0.17
Antihypertensive								
No	6	5537	1.08	7	5730	1.22	1.20(0.41, 3.50)	1.02(0.40, 2.61)
Yes	348	40,219	8.65	232	39,433	5.88	0.71(0.60, 0.83) ***	0.75(0.64, 0.89) ***
p for interaction								0.38
Antithrombotic								
No	40	24,106	1.66	15	24,127	0.62	0.40(0.22.0.71) **	0.42(0.24,0.77) **
Yes	314	21,650	14.5	224	21,036	10.7	0.76(0.64,0.90) **	0.80(0.68,0.95) **
P for interaction								0.02
Psychological therapy								
No	47	25,106	1.77	18	23,120	0.52	0.46(0.32, 0.61) **	0.43(0.26, 0.98) **
Yes	414	31,620	16.5	212	20,099	11.7	0.66(0.64, 0.90) **	0.79(0.66, 0.90) **
p for interaction								0.01

SHR = Subhazard Ratio; CI = Confidence Interval; PY = Persona Year; Rate ^#^, incidence rate, per 1000 person-years; Crude SHR, Relative Subhazard Ratio. ^a^ Adjusting for age, gender, comorbidities of hypertension, hyperlipidemia, chronic renal disease, gout, tobacco dependence-related, venous thrombosis, depression or substance-related, and medications of alpha glucosidase inhibitor (AGI), metformin, insulin, dipepidyl peptidase 4 (DPP-4) inhibitor, meglitinides, thiazolidinedione (TZD), sulphonylurea, statin, antidepressants, antihypertensive, antithrombotic and psychological therapy. Comorbidity ^‡^: Patients with any one of the comorbidities listed in Table 1 as the comorbidity group. The time between the diagnosis of DM and initial use of the GLP-1 RA; ^♥^ ≤49 ages: 1681 ± 7.9 days, ^♥♥^ 50–65 years: 1778 ± 8.6 days, ^♥♥♥^ ≥65 years: 2151 ± 9.6 days. The frequency of the GLP-1 RA use > 351 days in ≤ 49 ages, 50–65 years, ≥65 years were 56%, 45%, 10%, respectively; * *p* < 0.05, ** *p* < 0.01, *** *p* < 0.001.

**Table 4 biomedicines-12-00488-t004:** Incidence and adjusted subhazard ratio of stroke stratified by cumulative use day of GLP-1 R A therapy.

Medication Exposed	N	Event	Person-Year	Rate ^$^	Crude SHR (95% CI)	Adjusted SHR (95% CI) ^a^
GLP-1 RA	15,801	354	45,756	7.74		
No					1.00	1.00
Yes ^#^						
<85 days	4097	100	11,127	8.99	1.20(0.96, 1.50)	1.09(0.87, 1.37)
86–200 days	3719	59	8997	6.56	0.93(0.71, 1.23)	1.05(0.80, 1.39)
201–350 days	3961	41	9745	4.21	0.66(0.48, 0.91) *	0.73(0.52, 1.01)
≥351 days	4024	39	15,294	2.55	0.31(0.22, 0.43) ***	0.35(0.26, 0.49) ***

^#^ The cumulative use days per year are partitioned into 4 segments by quartile. Rate ^$^, incidence rate, per 1000 person-years. ^a^ Adjusting for age, gender, comorbidities of hypertension, hyperlipidemia, chronic renal disease, gout, tobacco dependence-related, venous thrombosis, depression or substance-related, and medications of alpha glucosidase inhibitor (AGI), metformin, insulin, dipepidyl peptidase 4 (DPP-4) inhibitor, meglitinides, thiazolidinedione (TZD), sulphonylurea, statin, antidepressants, antihypertensive, antithrombotic, and psychological therapy. Abbreviations: aSHR, adjusted subhazard ratio; * *p* < 0.05, *** *p* < 0.001.

**Table 5 biomedicines-12-00488-t005:** Overall estimated SHRs in CRD patients who used GLP-1 RA compared with CRD patients who did not use GLP-1 RA calculated by a time-dependent regression model.

	GLP-1 RA	
Variables	No (N = 15,801)	Yes (N = 15,801)
Stroke		
cSHR (95% CI)	1 (Reference)	0.36(0.28, 0.47) ***
aSHR (95% CI) ^a^	1 (Reference)	0.42(0.32, 0.55) ***
Ischemic stroke		
cSHR (95% CI)	1 (Reference)	0.51(0.41, 0.64) ***
aSHR (95% CI) ^a^	1 (Reference)	0.57(0.46, 0.72) ***
Hemorrhage stroke		
cSHR (95% CI)	1 (Reference)	0.46(0.34, 0.63) ***
aSHR (95% CI) ^a^	1 (Reference)	0.51(0.37, 0.71) ***

SHR = Subhazard Ratio; CI = Confidence Interval. ^a^ Adjusting for age, gender, comorbidities of hypertension, hyperlipidemia, chronic renal disease, gout, tobacco dependence-related, venous thrombosis, depression or substance-related, and medications of alpha glucosidase inhibitor (AGI), metformin, insulin, dipepidyl peptidase 4 (DPP-4) inhibitor, meglitinides, thiazolidinedione (TZD), sulphonylurea, statin, antidepressants, antihypertensive, antithrombotic and psychological therapy. *** *p* < 0.001.

## Data Availability

The dataset used in this study is held by the Taiwan Ministry of Health and Welfare (MOHW). The Ministry of Health and Welfare must approve our application to access this data. Any researcher interested in accessing this dataset can submit an application form to the Ministry of Health and Welfare requesting access (MOHW, Email: stcarolwu@mohw.gov.tw). Taiwan Ministry of Health and Welfare Address: No.488, Sec. 6, Zhongxiao E. Rd., Nangang Dist., Taipei City 115, Taiwan (R.O.C.). Phone: +886-2-8590-6848. All relevant data are within the paper.

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
