# Peer review of "Relationship between Glucagon-like Peptide-1 Receptor Agonists and Cardiovascular Disease in Chronic Respiratory Disease and Diabetes"

_biomedicines, 2024, doi:10.3390/biomedicines12030488_

Round 1
Reviewer 1 Report
Comments and Suggestions for Authors
This is a retropsective study aiming to assess the effectiveness of GLP-1 RAs on stroke and heart disease among patients having chronic respiratory disease and diabetes and underlying disease associated with COVID-19.
The main strengths of this manuscript are the main purpose and the large number of participants. However, it is very difficult for the readers to follow the manuscript, since there are several shortcomings and data interpretation issues that need to be addressed as outlined below:
1. Please use consistently either GLP1-RA or GLP-R1 (and explain this abbreviation, in case that it can be used)
2. Methods and materials:
- “Patients having CRD associated with COVID-19”: not clear the meaning of this population definition.
3. I would suggest that the inclusion criteria and the covariates analyzed in the multivariable models should be clearly formulated, since these parts of the manuscript are not clear for the readers.
4. It is also very critical that the authors have studied the detection of arrhythmia and not other common outcomes, including overall or cerebrovascular mortality.
5. Since the end of the study was on December 2019, why are the results presented almost 4 years later????
- Please also define the follow-up period for the patients included in this study.
6. The study was conducted only in China. The limited generalizability of the results should be discussed in the limitations of the study.
7. Minor comment: Typing and grammatic errors are detected in the manuscript. please revise accordingly.
Comments on the Quality of English Language
Moderate editing of English language required
Author Response
- Please use consistently either GLP1-RA or GLP-R1 (and explain this abbreviation, in case that it can be used)
Reply: Thank you for your comment. We revised the draft as your comments.
GLP-1 RAs: glucagon-like peptide-1 receptor agonists
- Methods and materials:
- “Patients having CRD associated with COVID-19”: not clear the meaning of this population definition.
Reply: Thank you for your comment. We revised the draft as your comments
We revised this sentence as below:
“Patients with CRD having impact on the outcomes of COVID-19:”
- I would suggest that the inclusion criteria and the covariates analyzed in the multivariable models should be clearly formulated, since these parts of the manuscript are not clear for the readers.
Reply: Thank you for your comment. We revised this paragraph as below:
Cohort identification
Patients with DM: newly-diagnosed patients were identified between 1998–2018 if they had (1) ≥1 inpatient record with a DM diagnosis (International Classification of Diseases, Ninth Revision, Clinical Modification (ICD-9-CM): 250; ICD-10-CM: E08–E13) or ≥2 outpatient (OPD) records with a DM diagnosis within the given years, or (2) ≥1 OPD record with a DM diagnosis and prescription records of glucose-lowering drugs within the given years.
Patients with CRD having impact on the outcomes of COVID-19: newly-diagnosed patients were identified between 1998–2018 if they had chronic respiratory diseases, such as COPD, asthma, lung cancer, bronchiectasis, tuberculosis, ILD, sleep apnea, sarcoidosis, pulmonary hypertension and chronic respiratory failure with ≥1 inpatient or ≥2 OPD records [1,2].
Definition of CD cohort: the initial cohort included patients with new-onset CRD and diabetes between (CD) 1998–2018.
Inclusion criteria: aged ≧20 years, if a patient had at least these two disease diagnosis requirements within 1 year of follow-up, a washout period of minimum 3 years was used to ensure that the patient was newly diagnosed with CD.
Exclusion criteria
We excluded patients: (1) whose sex were unknown, were not Taiwanese citizens, or younger than 20 years; (2) had no CRD-related prescription requirements or related management or procedure 1 year after the first CRD diagnosis; (3) had a disease history of CRD; (4) were diagnosed with DM before the first CRD diagnosis or no antidiabetic prescription statement or DM–related management or procedure, and (5) received GLP-1 RAs <3 times. The subsequent exclusion was used to increase the homogeneity of the study population.
Definition of GLP-1 RA use
The index date indicates when GLP-1 RA was first prescribed. The length of exposure to CD treatment with GLP-1 RA therapy was calculated as the time from the index date to the occurrence of the study outcomes or the final GLP-1 RA prescription during the observation period. A 90-day interval between prescription refills indicated therapy discontinuation. Patients were eliminated from the study when GLP-1 RA treatment was discontinued, a treatment switch occurred between the use of GLP-1 RA, or patients withdrew from the National Health Insurance program. Patient follow-up examinations continued until the investigated outcomes or end of the study (December 31, 2019), which ever occurred first.
- It is also very critical that the authors have studied the detection of arrhythmia and not other common outcomes, including overall or cerebrovascular mortality.
Reply: Thank you for your comment. We list these opoints as limitations.
Another limitation of this study is the limited generalizability of the findings. Although we used NHIRD data to obtain an adequate sample size, our results cannot be generalized to all GLP-I RA users because we recruited only patients living in Taiwan. Therefore ,large scale ,multi-center research warrant to confirm our results.
- Since the end of the study was on December 2019, why are the results presented almost 4 years later???? Please also define the follow-up period for the patients included in this study.
Reply: Thank you for your comment.
This is a cross-team study, including investigators from family medical departments, psychiatrists, chest physician, and they are from different hospitals. It is also a cross-level study including the residents, attending and statistics. We need to contact and integrate discussions and it will take a long time to reach a consensus.
Definition of GLP-1 RA use
----Patient follow-up examinations continued until the investigated outcomes or end of the study (December 31, 2019), whichever occurred first.
- The study was conducted only in China. The limited generalizability of the results should be discussed in the limitations of the study.
Reply: Thank you for your comment. We added this point in the limitations.
Another limitation of this study is the limited generalizability of the findings. Although we used NHIRD data to obtain an adequate sample size, our results cannot be generalized to all GLP-I RA users because we recruited only patients living in Republic of China (Taiwan). Therefore ,large scale ,multi-center research warrant to confirm our results
- Minor comment: Typing and grammatic errors are detected in the manuscript. please revise accordingly.
Reply: Thank you for your comment. We revised the draft as your comments.
- Comments on the Quality of English Language. Moderate editing of English language required.
Reply: Thank you for your comment. We revised the draft as your comments.

Reviewer 2 Report
Comments and Suggestions for Authors
The manuscript raises an interesting problem. The authors are aware of the limitations of the research since its retrospective character, which is a limitation itself. Overall, the study may be of scientific interest however some issues should be raised:
1. The authors should clearly state the aim of the study at the end of the Introduction section.
2. Methods: The study design should be clearly stated. A flow diagram would be better to present the study groups. How many patients were identified in a given year? Please clearly describe how the authors searched the Taiwan National Health Insurance Research Database (NHIRD).
3. Please divide the Results section into clear subsections with subsection headings to better follow the description of the results.
4. The p-value should be unified - with two or three decimal places.
5. What is the clinical significance of the findings of the study?
6. In reference number 1, the author's name should be in capital letters.
Comments on the Quality of English LanguageThere are some minor grammatical errors which need to be revised.
Author Response
- The authors should clearly state the aim of the study at the end of the Introduction section.
Reply: Thank you for your comment.
-----The aim of this study is that we would like to clarify the impact of the GLP-1 RA on the risk of the stroke and heart diseases among the CD cohort. Therefore, we addressed this topic based on a time-dependent analysis after propensity matching of the study population.
- Methods: The study design should be clearly stated. A flow diagram would be better to present the study groups. How many patients were identified in a given year? Please clearly describe how the authors searched the Taiwan National Health Insurance Research Database (NHIRD).
Reply: Thank you for your comment.
- Please divide the Results section into clear subsections with subsection headings to better follow the description of the results.
Reply: Thank you for your comment.
Results
The demographic characteristics of the study population are presented in Table 1.
The Table 2 display the incidence of stroke and heart disease by Cox model measured hazards ratio according to medication status such as the time between the diagnoses of the DM and the initial use of the GLP-1 RA.
The Table 3 display the comorbidities, medications and psychological therapy might interfere the effect of the GLP-1 RA use for the stroke and heart diseases.
Table 4 display the incidence and adjusted subhazard ratio of stroke after stratification of the cumulative use day of GLP-1 R A therapy.
The Table 5 display the time-dependent regression for the GLP-1 RA use.
The Figure 1, 2 and 3 display the stroke and arrhythmia cumulative incidence curve.
- The p-value should be unified - with two or three decimal places.
Reply: Thank you for your comment.
Table 1. Distributions of Demographic and Clinical Comorbid Status in GLP-1 RA among diabetes with chronic respiratory disease (CRD) by Propensity Score Matching.
|
|
GLP-1 RA |
|
|||
|
|
No N=15801 |
Yes N=15801 |
|
||
|
|
n |
% |
n |
% |
p-value |
|
Age, years |
|
|
|
|
0.03* |
|
≦49 |
6153 |
38.9 |
6373 |
40.3 |
|
|
50-64 |
6640 |
42.0 |
6521 |
41.3 |
|
|
≧65 |
3008 |
19.0 |
2907 |
18.4 |
|
|
Mean±SD a |
54.7 |
12.7 |
52.6 |
12.8 |
0.00* |
|
Gender |
|
|
|
|
0.57 |
|
Women |
8481 |
53.7 |
8531 |
54.0 |
|
|
Men |
7320 |
46.3 |
7270 |
46.0 |
|
|
Comorbidity |
|
|
|
|
|
|
Hypertension |
11013 |
69.7 |
10957 |
69.3 |
0.49 |
|
Hyperlipidemia |
14388 |
91.7 |
14376 |
91.0 |
0.03* |
|
Chronic renal disease |
1861 |
11.8 |
1977 |
12.5 |
0.05 |
|
Gout |
2430 |
15.4 |
2479 |
15.7 |
0.45 |
|
Tobacco dependence -related |
341 |
2.16 |
372 |
2.35 |
0.24 |
|
Venous thrombosis |
19 |
0.12 |
22 |
0.14 |
0.64 |
|
Depression or substance related disease |
1927 |
12.2 |
1990 |
12.6 |
0.28 |
|
Medications |
|
|
|
|
|
|
AGI |
8856 |
56.1 |
8926 |
56.5 |
0.43 |
|
Metformin |
15737 |
99.6 |
15660 |
99.1 |
0.00* |
|
Insulin |
13008 |
82.3 |
12972 |
82.1 |
0.60 |
|
DPP-4 inhibitor |
13973 |
88.4 |
13873 |
87.8 |
0.08 |
|
Meglitinides |
4878 |
30.9 |
5072 |
32.1 |
0.02* |
|
TZD |
9872 |
62.5 |
9813 |
62.1 |
0.49 |
|
Sulphonylurea |
14731 |
93.2 |
14503 |
91.8 |
0.00* |
|
Statin |
13727 |
86.9 |
13637 |
86.3 |
0.14 |
|
Antidepressants |
5311 |
33.6 |
5494 |
34.8 |
0.03* |
|
Antihypertensive |
13813 |
87.4 |
13725 |
86.9 |
0.14 |
|
Antithrombotic |
7428 |
47.0 |
7312 |
46.3 |
0.19 |
|
Psychotherapy |
5300 |
33.5 |
5420 |
34.3 |
0.06 |
|
Chi-square test, a t-test Abbreviations: *p value<0.05. |
|||||
AGI: alpha glucosidase inhibitor, DPP-4 inhibitor: dipepidyl peptidase 4 inhibitor, TZD: thiazolidinedione.
- What is the clinical significance of the findings of the study?
Reply: Thank you for your comment.
Both a shorter lag time use of the GLP-1 RA and longer use of GLP-1 RA were associated with a decreased risk of ischemic or hemorrhagic stroke in CD cohort. The GLP-1 RA use in the early stage with optimal time use in the CD cohort may avoid the risk of stroke . These findings suggest implications for clinical practice and further research.
- In reference number 1, the author's name should be in capital letters.
Reply: Thank you for your comment.
Boubaker, N.; Louhaichi, S.; Khalfallah, I.; Belkhir, S.; Ferchichi, M.; Ammar, J.; Hamdi, B.; Hamzaoui, A. Prevalence and impact of chronic respiratory disease in moderate to severe COVID-19 outcomes. European Respiratory Journal. 2021, 58, PA301.
- Comments on the Quality of English Language.
There are some minor grammatical errors which need to be revised.
Reply: Thank you for your comment. We revised the draft as your comments.
